# An LNA-amide modification that enhances the cell uptake and activity of phosphorothioate exon-skipping oligonucleotides

Ysobel R. Baker [1], Cameron Thorpe [1], Jinfeng Chen[1], Laura M. Poller [1], Lina Cox [1], Pawan Kumar[1], Wooi F. Lim [2], Lillian Lie[1], Graham McClorey[2], Sven Epple [1], Daniel Singleton[3], Michael A. McDonough [1], Jack S. Hardwick [1], Kirsten E. Christensen [1], Matthew J. A. Wood [2], James P. Hall [4], Afaf H. El-Sagheer [1,5] & Tom Brown [1✉]

Oligonucleotides that target mRNA have great promise as therapeutic agents for life-threatening conditions but suffer from poor bioavailability, hence high cost. As currently untreatable diseases come within the reach of oligonucleotide therapies, new analogues are urgently needed to address this. With this in mind we describe reduced-charge oligonucleotides containing artificial LNA-amide linkages with improved gymnotic cell uptake, RNA affinity, stability and potency. To construct such oligonucleotides, five LNA-amide monomers (A, T, C, 5mC and G), where the 3′-OH is replaced by an ethanoic acid group, are synthesised in good yield and used in solid-phase oligonucleotide synthesis to form amide linkages with high efficiency. The artificial backbone causes minimal structural deviation to the DNA:RNA duplex. These studies indicate that splice-switching oligonucleotides containing LNA-amide linkages and phosphorothioates display improved activity relative to oligonucleotides lacking amides, highlighting the therapeutic potential of this technology.

[1] Department of Chemistry, University of Oxford, Chemistry Research Laboratory, 12 Mansfield Road, Oxford OX1 3TA, UK. [2] Department of Paediatrics, University of Oxford, LGC building, South Parks Road, Oxford OX1 3QX, UK. [3] ATDBio Ltd, School of Chemistry, University of Southampton, Highfield, Southampton SO17 1BJ, UK. [4] Department of Pharmacy, Chemistry and Pharmacy Building, University of Reading, Whiteknights, Reading RG6 6AD, UK. [5] Department of Science and Mathematics, Suez University, Faculty of Petroleum and Mining Engineering, Suez 43721, Egypt. ✉email: tom.brown@chem.ox.ac.uk

Antisense oligonucleotides (ASOs) are synthetic nucleic acid analogues that bind to RNA to regulate gene expression and interfere with protein synthesis[1,2]. Their modes of action include steric blocking of translation[3], recruitment of RNase-H for degradation of mRNA[4], modulation of pre-mRNA splicing[5–7] and siRNA-mediated gene silencing[8]. Therapeutic oligonucleotides in general have shown great promise as precision medicines, and have the potential to cure cancers, genetic disorders, and infectious diseases. They are particularly attractive because there are far more RNA targets than conventional protein targets in currently undruggable diseases[9]. Other advantages include their simple design criteria based on Watson–Crick base pairing and high target specificity. However, although the ASO concept was first demonstrated as far back as 1978[10], ASOs have only recently started to deliver on their initial promise. Fifteen oligonucleotides have been approved for clinical use (twelve since 2016) indicating their increasing therapeutic value[11]. Until recently all of these were for rare life-limiting diseases, but the field just received a major boost with the EMA and NICE approval of the Novartis siRNA drug Inclisiran (Leqvio) for lowering LDL cholesterol levels in patients with hypercholesterolaemia[12]. Unlike previous oligonucleotide therapeutics, Inclisiran is used to treat a very common chronic disease. Although a major triumph, and an insight into what the future could hold for oligonucleotide therapeutics, Inclisiran is a special case where targeted delivery to the liver is required. Other organs are more difficult to penetrate, and inefficient biodistribution, poor cellular delivery, and toxicity prevent the wider adoption of antisense technology. Given these limitations, new oligonucleotide analogues are urgently needed to advance the field.

To achieve a therapeutic response an oligonucleotide must be stable in vivo, bind to its target RNA with high selectivity and affinity, and display good pharmacokinetic properties[2]. Unmodified oligonucleotides are rapidly digested by nucleases in cells and are therefore completely unsuitable. Modifications designed to provide nuclease resistance[13] include 2′-OMe, 2′-O-(2-methoxyethyl), 2′-fluoro sugars[14–16] and phosphorothioate (PS) backbones (Fig. 1a)[15,17]. The PS group improves cell uptake but reduces RNA target affinity, which can be restored by the 2′-sugar substituents[2]. Whilst these modifications have led to oligonucleotide therapies[11], limited efficacy, off-target effects and toxicity remain obstacles to wider adoption in the clinic[18]. Poor uptake into cells is particularly troublesome; less than 1% of an administered oligonucleotide typically reaches its target[19]. Reducing the net anionic charge of oligonucleotides by partial or complete replacement of the phosphodiester backbone with neutral linkages would seem to be a viable strategy for increasing cell permeability and nuclease resistance[13,20]. A clinically relevant example[21] is the phosphorodiamidate-morpholino backbone (PMO) (Fig. 1a), but limitations in solid-phase syntheses complicate the assembly of charge-reduced PMO–DNA chimaeras. Most other uncharged backbones that have been studied reduce the affinity and selectivity of therapeutic oligonucleotides towards their RNA targets[13]. However, we felt that the 'reduced charge' principle warranted detailed investigation. Oligonucleotides containing the artificial amide backbone AM1 (Fig. 1b) are of interest as they form moderately stable duplexes with complementary RNA[22–27]. Despite this, the required monomers have not been synthesised for all four canonical nucleobase analogues[28]. Inspired by the extremely high target affinity of locked nucleic acids (LNA) (Fig. 1b) developed in the 1990's[29–32], we hypothesised that conformationally locking the nucleosides surrounding the AM1 linkage would produce reduced-charge oligonucleotides (ONs) with higher target affinity, and in combination with PS, linkages might yield ASOs with superior cell uptake and in vivo stability. Although attaching LNA sugars directly to other non-phosphorus-based charge-neutral backbones has so far failed to produce analogues with increased duplex stability[33–38] we reasoned that AM1, being a particularly close analogue of the phosphodiester backbone[39], might behave differently (Fig. 1c).

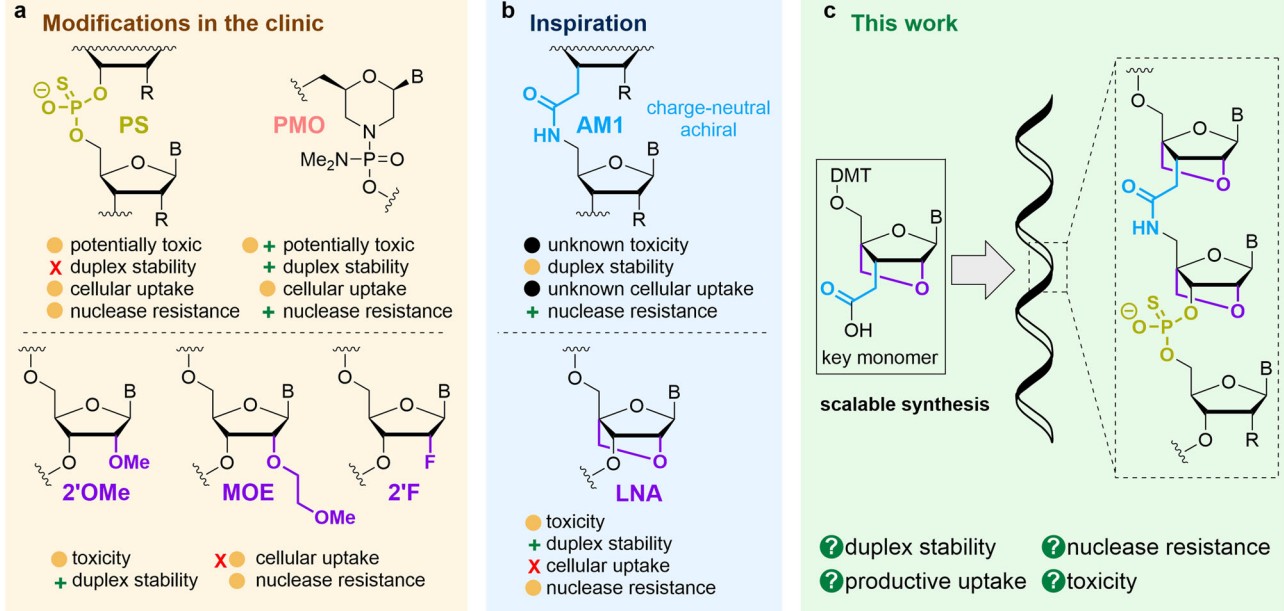

**Fig. 1 Therapeutic oligonucleotide modifications and our strategy for combining these. a** Existing therapeutic oligonucleotide modifications and their properties. Traffic light coding system: green plus = good; amber circle = intermediate; red cross = poor. **b** Oligonucleotide modifications that have not yet reached the clinic **c** Development of a new antisense backbone chemistry with improved target affinity and cellular activity. The LNA-amide linkage is achiral and can be prepared by solid-phase synthesis methods that are high yielding with the potential to be automated. PS phosphorothioate, PMO phosphorodiamidate-morpholino, MOE methoxyethyl, AM1 amide, LNA locked nucleic acid, DMT 4, 4′-dimethoxytrityl.

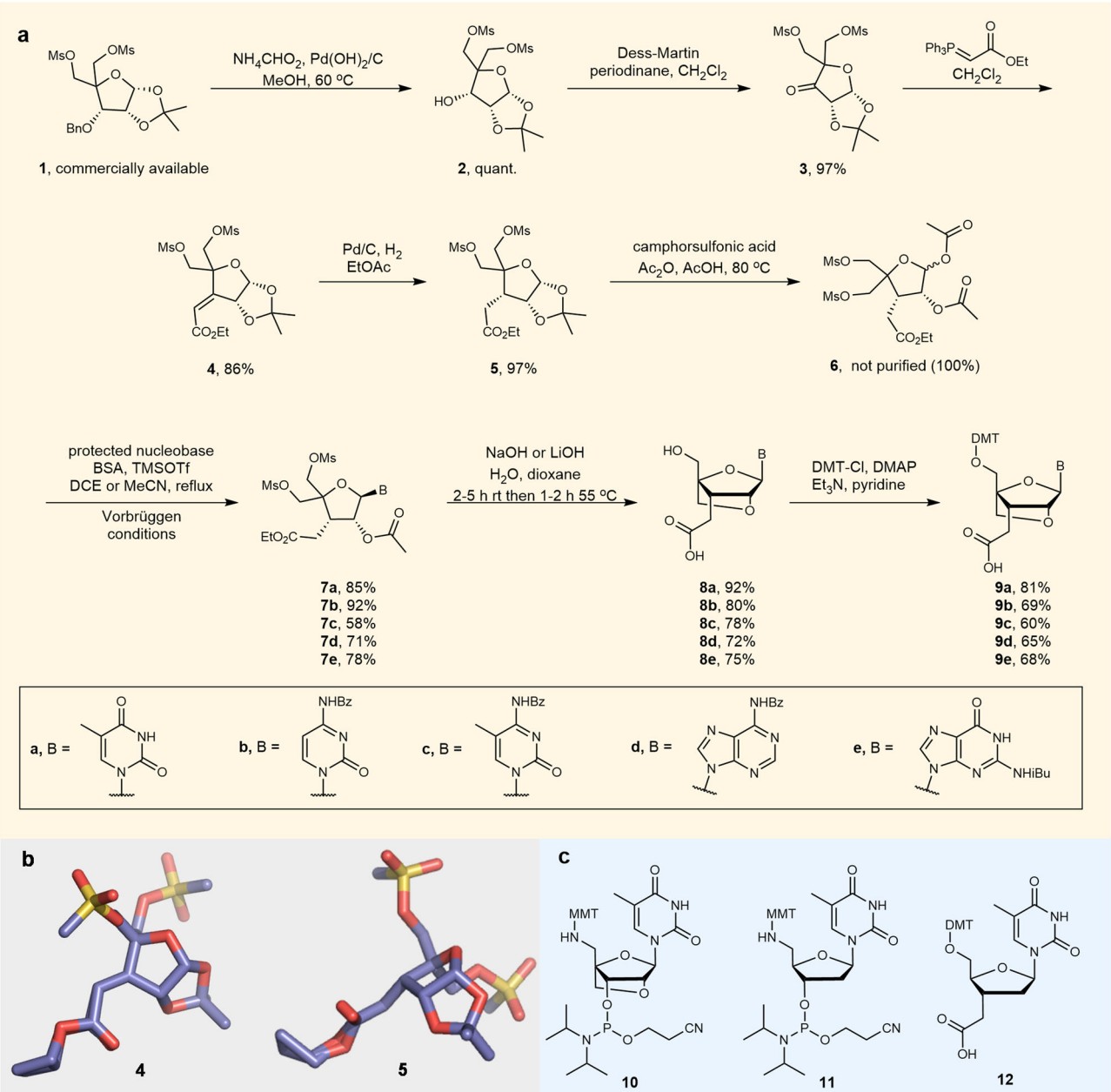

**Fig. 2 LNA-ethanoic acids and other monomers used in this study. a** Synthesis of the DMT-protected LNA-ethanoic acids **9a**–**e**. **b** X-ray crystal structures confirming the (E)-configuration in **4** and the stereochemistry at the 3′-carbon in **5** (details in 'Small molecule crystallography' section in the Supplementary Information). **c** Phosphoramidites **10**[47] and **11** (commercially available) and DNA-ethanoic acid monomer **12**[48,49] are used to synthesise oligonucleotides. BSA bis(trimethylsilyl)acetamide, TMSOTf trimethylsilyl trifluoromethanesulfonate, DMAP 4-dimethylaminopyridine, DMT 4,4′-dimethoxytrityl, MMT 4-monomethoxytrityl.

Here, we describe the synthesis of oligonucleotides containing LNA-amide linkages and show that these can be combined with phosphodiester or phosphorothioate backbones. We demonstrate the excellent duplex stabilisation of the LNA-amide linkages, mismatch discrimination and extreme resistance to nucleases. Preliminary studies show that LNA-amide ONs containing PS backbones have improved cellular uptake and are highly effective in exon-skipping assays.

## Results and discussion
**LNA-amide monomer synthesis is efficient and scalable.** Efficient synthesis of chemically modified oligonucleotides is essential to fuel fundamental studies and therapeutic applications. We

have devised a strategy which enables the straightforward assembly of oligonucleotides containing LNA-flanked amides which could be easily automated and scaled up. A maximum of eight monomers are required to make any sequence that contains non-contiguous amide linkages (four carboxylic acids and four amines).

Attaching the required ethanoic acid moiety to the 3′-carbon of the LNA sugar is challenging as it requires the removal of the 3′-oxygen and the formation of a C-C bond. However, we were able to produce the required 5′-dimethoxytrityl (DMT)-protected 3′-ethanoic acid LNA-monomers in 8 steps from **1**, the same number as for conventional LNA phosphoramidites[40] (Fig. 2a). Moreover, we did this with minimal chromatography. We first

built the sugar with the C3′-ester **5**, before the addition of the nucleobase. This approach avoids complications associated with forming a C-C bond at the 3′-side of a nucleoside using the Barton-McCombie reaction[23–25,41] or the hydrogenation step if a Wittig reaction is used[42,43], which would ultimately limit the range of heterocyclic bases that can be added. Key intermediate **5** was readily prepared on a multiple gram scale in 81% overall yield from commercially available **1** without the need for chromatographic separation as follows. Hydrogenolysis of compound **1** afforded alcohol **2** which was subsequently oxidised to give compound **3**. Olefination of **3** with (carbethoxymethylene) triphenylphosphorane (Wittig reaction) selectively yielded **4** as the (E)-stereoisomer (Fig. 2b). Catalytic hydrogenation of **4** using Pd/C and $H_2$ gave **5** as a single stereoisomer (Fig. 2b). This stereoselectivity was predicted because the 1,2-O-isopropylidene groups on the α-face of furanosyl carbohydrate derivatives direct incoming $H_2$ to the β-face[44]. Key intermediate **5** was then converted to the 1,2-di-O-acetate glycosyl donor **6** following a procedure reported by Arzel et al.[45], avoiding the formation of a lactone which occurred when the acetonide was cleaved in the presence of water.

The pathways to each monomer then diverged, with Vorbrüggen conditions[40,46] utilised for the addition of the nucleobases to access **7a–e**. Subsequent simultaneous unmasking of the 3′-carboxyl and 2′-hydroxyl groups by treatment with hydroxide, followed by cyclisation to form the 2′-4′-oxymethylene bridge, then 5′-mesyl deprotection, gave the hydroxy-LNA-acid compounds **8a–e**. The progress of the reaction was rapid; mesyl deprotection using hydroxide ion conventionally requires several days under reflux conditions. We postulate that the acceleration in rate is due to neighbouring group participation whereby the carboxylate anion displaces the 5′-mesyl group, forming a lactone that is subsequently opened by hydrolysis (Supplementary Fig. 1). Finally, we treated the resulting hydroxy-acids **8a–e** with 4,4′-dimethoxytrityl chloride (DMT-Cl) in pyridine to give the DMT-protected LNA analogue nucleosides **9a–e**. Using this strategy we were able to access all four canonical nucleoside analogues along with the 5-methylcytidine monomer which is used in place of cytidine in antisense experiments to increase target affinity and improve other therapeutic properties[18]. Additionally, we required 5′-MMT-amino LNA phosphoramidite **10**. Whilst this had been previously synthesised[47] we chose to develop a more efficient route (Supplementary Fig. 2). Commercially available 5′-MMT-amino dT **11** and 5′-DMT thymidine-3′-ethanoic acid **12**[48,49] (Fig. 2c) were also required to enable us to make oligonucleotides to compare the properties of DNA-amide with those of LNA-amide.

**Chimeric oligonucleotides can be synthesised in high purity**. Our oligonucleotide synthesis strategy is shown in Fig. 3. A phosphoramidite monomer with an MMT-protected 5′-amino group, either LNA **10**[40] or deoxythymidyl **11**, is added to the oligonucleotide, and the amine is deprotected using trichloroethanoic acid. An LNA-acid (or DNA-acid[50]) monomer is coupled to the free amine using PyBOP activating agent in the presence of a non-nucleophilic base (N-methylmorpholine) to form the amide bond. Oligonucleotide synthesis is then resumed, starting with the removal of the DMT group. The process is repeated to install multiple non-contiguous amides in the same oligonucleotide. To demonstrate this, DMT-protected LNA acids **9a–e**, phosphoramidites **10** and **11**, and DNA-acid **12**[48,49] (Fig. 2), were used to synthesise several oligonucleotides, some of which contain multiple additions of LNA-amide, 2′-OMe sugars and PS linkages (Supplementary Table 1). In all cases, we obtained the oligonucleotides in high purity. High-performance liquid chromatography (HPLC) and mass spectrometry

demonstrated the high incorporation efficiency for the DMT-protected LNA acids **9a–e** (Supplementary Figs. 3–7).

**LNA sugars stabilise duplexes containing the amide backbone**. To evaluate the ability of the LNA-amide combination to bind to complementary RNA with high affinity we synthesised a series of 13-mer ONs with a central amide with a T-T sequence. These constructs were composed of either no LNA (ON1$^{DNA-Am-DNA}$), an LNA 5′ to the amide (ON2$^{LNA-Am-DNA}$), an LNA 3′ to the amide (ON3$^{DNA-Am-LNA}$), or LNA on both sides of the amide (ON4$^{LNA-Am-LNA}$, Fig. 4a). Controls without amide (ON5$^{LNA-LNA}$ and ON6$^{DNAcontrol}$) were also made. We compared duplex denaturation temperatures ($T_m$s) after hybridisation with DNA and RNA complementary strands (Fig. 4b, Supplementary Table 2 and Supplementary Figs. 8, 9). ON2$^{LNA-Am-DNA}$ produced a significant increase in DNA:RNA hybrid stability, (+3.0 °C) compared to the unmodified ON6$^{DNAcontrol}$, and an increase of +3.4 C compared to 'amide only' ON1$^{DNA-Am-DNA}$. Importantly, ON4$^{LNA-Am-LNA}$, in which the amide is surrounded by LNA sugars, gave the greatest increase in stability (+5.1 °C). It is noteworthy that ON2$^{LNA-Am-DNA}$ and ON4$^{LNA-Am-LNA}$ provide the first examples of duplex stabilisation by an LNA sugar attached to a non-phosphorus artificial DNA backbone. The RNA target selectivity of LNA-amide-containing ONs was excellent; a single mismatched base pair greatly reduced duplex stability, in some cases by >14 °C (Supplementary Table 2 and Supplementary Figs. 10–13) In summary, an amide linkage flanked by LNA on both sides gives strong DNA:RNA duplex stabilisation and good mismatch discrimination.

In duplexes with DNA targets, ONs with all combinations of LNA and DNA sugars around the amide linkage were slightly destabilising (between −0.1 to −2.6 °C), indicating the selectivity of the amide linkage for complementary RNA. The stabilisation induced by the LNA-amide combination is cumulative and general (Fig. 4c, Supplementary Table 3 and Supplementary Figs. 14–17). In a biologically relevant sequence context, four LNA-amides increase duplex stability against complementary RNA by an impressive 13.0 °C compared to only 5.1 °C for the DNA target. This large difference is important when developing oligonucleotides to interact with RNA. Oligonucleotides for in vivo studies usually contain 2′-OMe modified sugars and/or phosphorothioate backbones to prevent degradation by nucleases. In such oligonucleotides, the combination of LNA and amide also greatly increase duplex stability (Supplementary Table 3).

**Combining LNA and amide provides strong nuclease resistance**. Therapeutic oligonucleotides must remain stable in cells for prolonged periods to remain active. To evaluate whether the combination of LNA and amide confers greater nuclease resistance than LNA alone, we incubated unmodified DNA (ON15$^{DNA/17PO}$) and DNA with four LNA-amide linkages (ON14$^{DNA/4LAL/13PO}$) in a 1:1 mixture of phosphate-buffered saline (PBS) and foetal bovine serum (FBS) to mimic the in vivo environment and compared it to the equivalent construct with LNA but without amide linkages (ON25$^{DNA/8LNA/17PO}$, Supplementary Fig. 18). The results show that the combination of LNA and amide confers extreme resistance to nucleases. Both the oligonucleotides lacking amide linkages had partially degraded within 1 h whereas ON14$^{DNA/4LAL/13PO}$ remained intact after 8 h. Interestingly ON14 shows stability at its 3′-end, even though this region has an unmodified sugar-phosphate trimer. The 3′-terminal pentamer region, however, is highly modified. It has a reduced charge due to the amide linkage and also contains two LNA sugars. It may therefore not be recognised by nucleases. This enhanced stability further illustrates

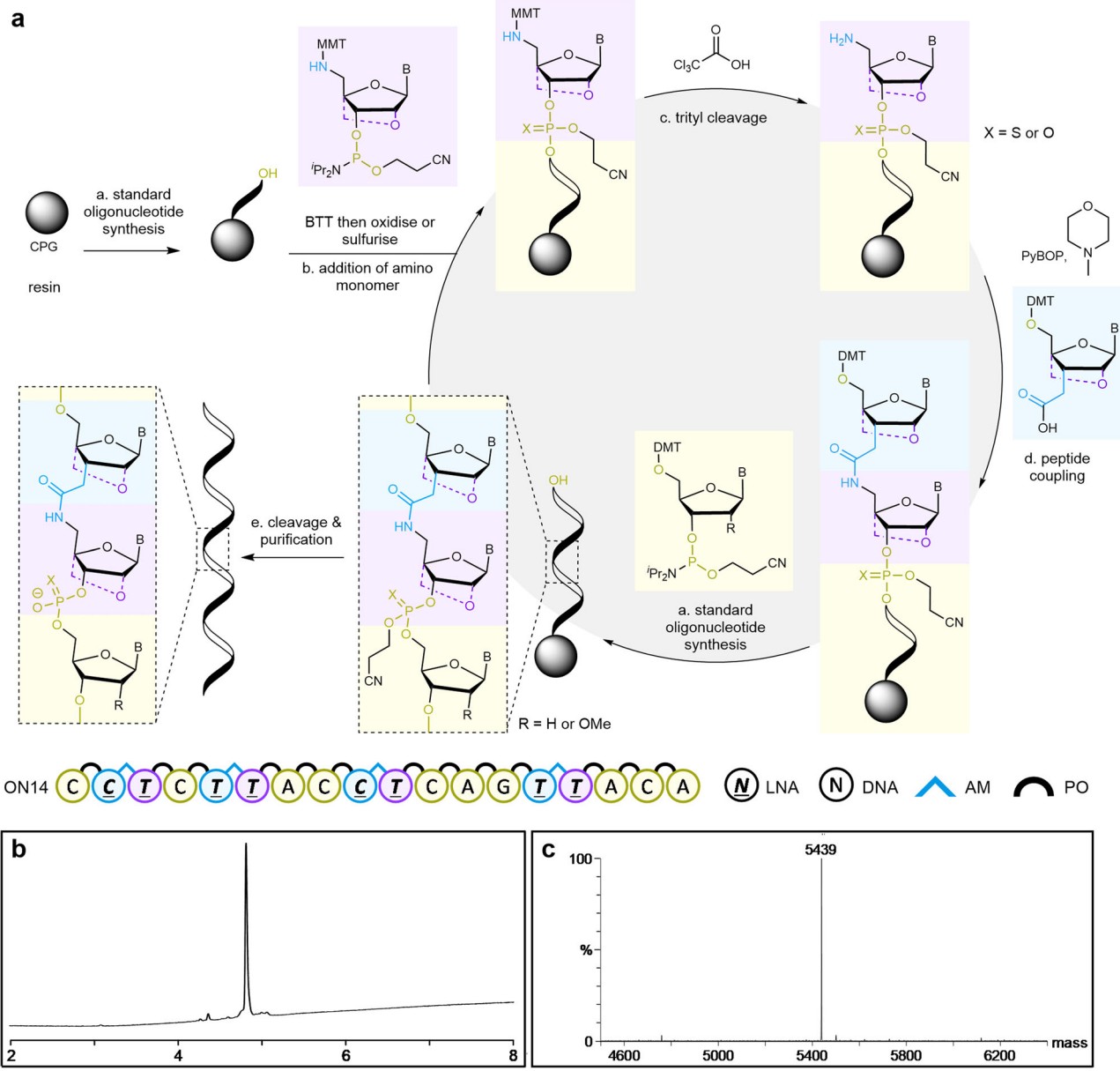

**Fig. 3 Synthesis of (LNA)-amide-phosphodiester chimeric oligonucleotides. a** Efficient solid-phase synthesis of amide-phosphodiester chimaeras. Dashed lines indicate the presence or absence of 2′-4′-methylene bridge. PyBOP benzotriazol-1-yloxytripyrrolidinophosphonium hexafluorophosphate. **b** Reversed-phase HPLC chromatogram. X-axis = time in minutes. y-axis = absorbance at 260 nm. **c** Electrospray ES- mass spectrum of a crude LNA-amide chimeric oligonucleotide (ON14). X-axis = mass Y-axis = relative intensity. Required mass = 5438.7 Da, Found mass = 5439.0 Da.

the advantages of removing charge and including modified sugars in antisense oligonucleotides.

**X-ray structures of the LNA-amide modification in DNA-RNA hybrids**. We solved several X-ray structures to determine the effects of LNA and amide modifications on duplex structure and conformation. These are the first crystal structures of DNA:RNA hybrids that contain amide linkages. An amide-modified RNA:RNA duplex was analysed previously, but this had amides in *both* strands surrounded by multiple mismatched base pairs which cannot exist outside the solid state at ambient temperature[27]. The sequence of the modified DNA:RNA hybrid duplexes (Supplementary Table 4) was based on the corresponding unmodified version (d-CTTTTCTTTG/rCAAAGAAAAG)[51] (the location of the amide is underlined). Good quality crystals

diffracting between 2.5–2.8 Å resolution were obtained for the DNA:RNA hybrid in which the DNA strand contains an amide linkage flanked by DNA on both sides, LNA on both sides and with LNA only on the 5′-side (Supplementary Fig. 19). The unmodified DNA:RNA duplex was also studied. The data collection and refinement statistics are given in Supplementary Table 5. Electron density maps at the modification position for the four nucleic acid crystal structures reported in this manuscript are given in Supplementary Fig. 20. The hybrids with amide and LNA-amide backbones (Fig. 5a) are structurally very similar to the unmodified duplex (all-atom RMSD 0.4 Å) as shown by their superimposition (Fig. 5a and Supplementary Fig. 21). All structures adopt the A-conformation with sugar puckers clustering around C3′-*endo* (Supplementary Fig. 22). As expected, all duplexes are stabilised by canonical Watson–Crick base pairs, indicating that the thermodynamic improvements due to the

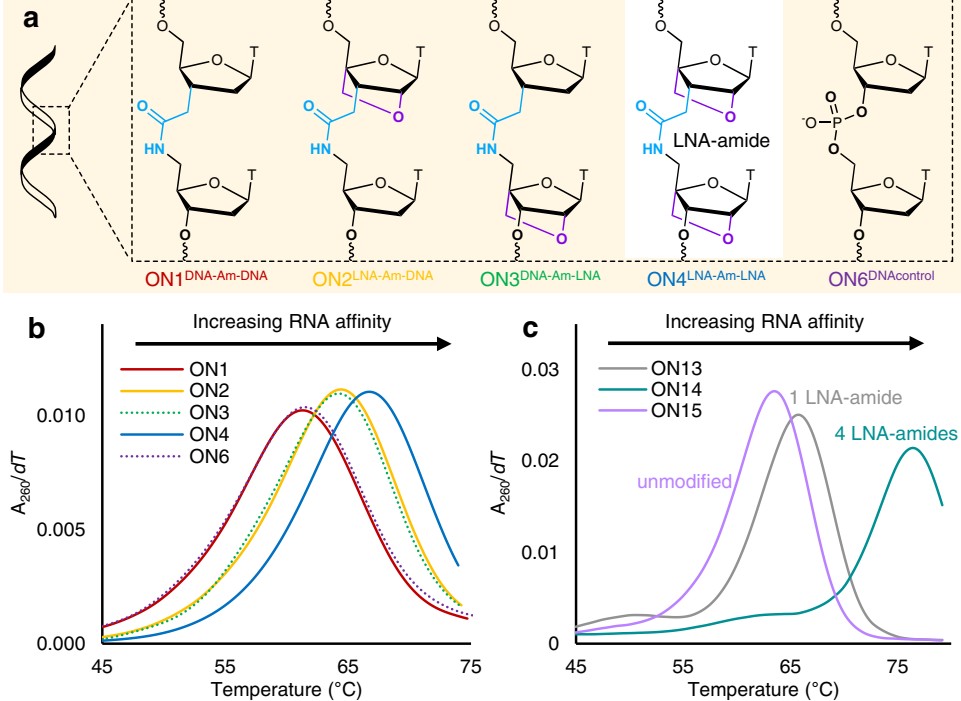

**Fig. 4 Combining LNA and AM1 increase affinity for complementary RNA. a** The structures of each of the LNA AM1 combinations studied. Sequence d-CGACGC**TT**GCAGC (where the underlined region indicates the modification position) **b** UV-melting curve derivatives of the ONs from panel **a** in duplexes with complementary RNA. **c** UV-melting curve derivatives demonstrate that the stabilisation conferred by the LNA-amide linkage against complementary RNA is general and cumulative. Conditions: 10 mM phosphate buffer, 200 mM NaCl, pH 7.0. Sequences in Supplementary Table 1.

LNA-amide backbones are not due to unusual changes in hydrogen bonding interactions. In agreement with the DNA:RNA hybrid NMR structure by Rosners[22] in which the DNA strand contained multiple amides, our X-ray studies indicate that the amide linkage is a close mimic of the phosphodiester backbone (Fig. 5b). Both are four-atom linkages, hence similar in length, and the amide carbonyl is orientated in the same direction as one of the phosphodiester P-O bonds.

In Fig. 5c, the structures of all amide backbones are overlaid to assess the effects of the LNA modifications. Between each structure, the orientation of the backbone is consistent, directing the amide oxygen into the major groove. Other atomic positions of the backbones also show close similarity, and the presence of 3′-LNA causes no significant distortion. 5′-LNA does however cause some structural displacement; the 5′-sugars in the LNA-amide-DNA and LNA-amide-LNA structures are shifted slightly outwards compared to the DNA-amide-DNA and unmodified strands. Despite this, the positioning of the amide backbone remains consistent between each structure. The amide adopts the expected *trans*-conformation, and LNA on the 5′-side of the amide has little effect on backbone torsion angles (Fig. 5c and Supplementary Fig. 23). In summary, combined LNA and amide modifications have minimal effect on the duplex structure, and are excellent mimics of natural phosphodiesters.

**Combining LNA-amide and PS enhances gymnotic delivery.** In a preliminary study, we have evaluated the biological activity of the LNA-amide combination using the HeLa pLuc/705 cell line[52] that carries a luciferase-encoding gene interrupted by a mutated ß-globin intron[52]. This mutation creates a 5′-splice site which activates a cryptic 3′-splice site, resulting in incorrect mRNA splicing and the production of non-functional luciferase. An oligonucleotide that hybridises to the mutant 5′-splice site prevents incorporation of the aberrant intron, restoring the pre-

mRNA splicing to produce functional luciferase, which is quantified by luminometry. Oligonucleotides complementary to this aberrant splice site were synthesised with combinations of different modifications to determine their individual effects (Supplementary Table 3). ON14$^{DNA/4LAL/13PO}$, ON16$^{2'OMe/4LAL/13PO}$, and ON18$^{2'OMe/4LAL/13PS}$ were designed to evaluate LNA-amide with the DNA, 2′-OMe/phosphodiester, and 2′-OMe/phosphorothioate backbones respectively, and ONs 14 and 16 were included to determine the degree to which LNA-amide influences delivery, activity and toxicity in the absence of PS linkages. LNA and amide linkages are incompatible with RNase-H so there is no risk of ON14 or ON16 inadvertently destroying the RNA target[2]. Three controls were included: ON20$^{2'OMe/17PS}$ (which represents the gold standard in the assay) to determine whether LNA-amide enhances biological activity[52], ON17$^{2'OMe/17PO}$ to evaluate the effects of the PS linkage independently of LNA or amide linkages, and ON19$^{2'OMe/8LNA/17PS}$ with LNA sugars but no amide linkages to determine the effects of the enhanced duplex stability caused by LNA. A scrambled control with a 2′-OMe/PS backbone was also included to determine off-target effects (ON31$^{2'OMe/17PS\ scrambled}$, Fig. 6).

To compare biological activity independent of cell uptake, Lipofectamine 2000 (LF2000), a cationic liposome transfection/delivery reagent, was used. All three target-complementary PS-ONs were active in the assay (ON20$^{2'OMe/17PS}$, ON18$^{2'OMe/4LAL/13PS}$ and ON19$^{2'OMe/8LNA/17PS}$), whereas all the PO-ONs (ON14$^{DNA/4LAL/13PO}$, ON16$^{2'OMe/4LAL/13PO}$ and ON17$^{2'OMe/17PO}$) were inactive at 100 nM (Fig. 6a). Hence, in agreement with previous studies, phosphorothioate modification in a target-complementary sequence is necessary for splice-switching activity. This could result from the PS groups enhancing nuclear enrichment[53] of the oligonucleotides, and/or recruiting ILF2/3 to the RNA transcript[54]. Notably, the addition of the amide linkage significantly improved the splice-switching activity of 2′-OMe/PS-ONs at the lower

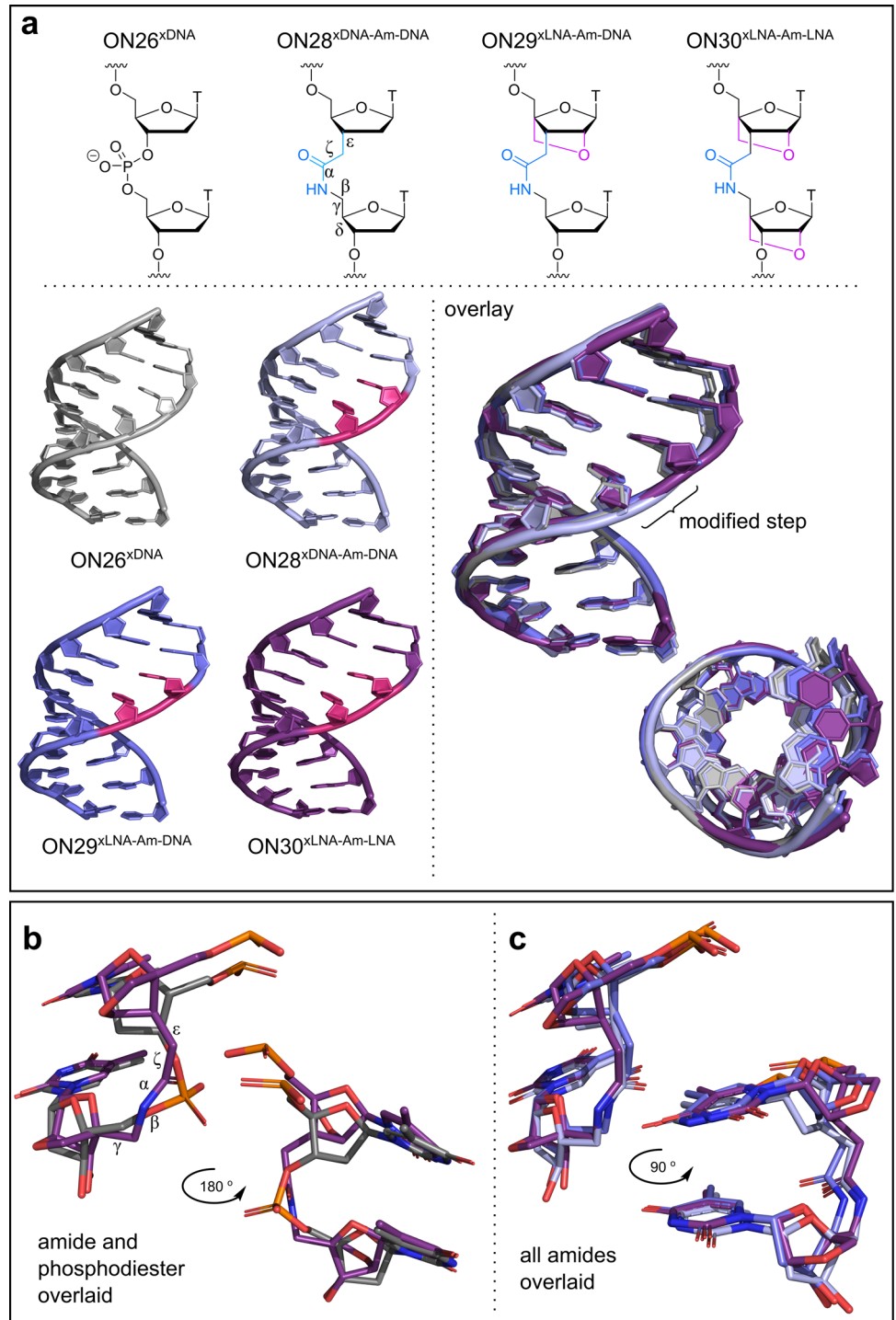

**Fig. 5 X-ray structures of amide and LNA-amide-modified DNA:RNA duplexes. a** Structural identity of amide and LNA-amide modifications and the torsion angles of the amide backbone (5′ε → 3′δ). Pink steps show the modification position (left) and the overlay of all structures shows clear similarities (right). **b** Backbone overlay comparing the LNA-amide-LNA step in ON30[xLNA-Am-LNA] (purple) with the phosphodiester in ON26[xDNA] (grey). **c** Overlay of all amide backbones with or without LNA modifications (light blue = ON28[xDNA-Am-DNA], dark blue = ON29[xLNA-Am-DNA], purple = ON30[xLNA-Am-LNA]).

concentrations (6.25 nM and 12.5 nM), probably due to improved target affinity (Fig. 6b). Next, we compared the naked (gymnotic) uptake of the ONs. These conditions more closely represent in vivo applications where transfection agents such as LF2000 cannot be used. We seeded cells at low confluency, added the oligonucleotides in fresh media after 16 h, and measured luciferase activity after a further 96 h. The presence of just four LNA-amides (ON18[2′OMe/4LAL/13PS]) significantly increased activity

in a dose-dependent manner compared to ON20[2′OMe/17PS] (Fig. 6c). Greater than fivefold increase in activity was observed for gymnotic delivery compared to a maximum of threefold increase for the LF2000-mediated transfection. This suggests that synergy between the PS and LNA-amide modifications leads to enhanced productive delivery into cells. The improved therapeutic properties of LNA-amide/PS oligonucleotides observed in these preliminary studies are possibly due to a combination of reduced charge from the

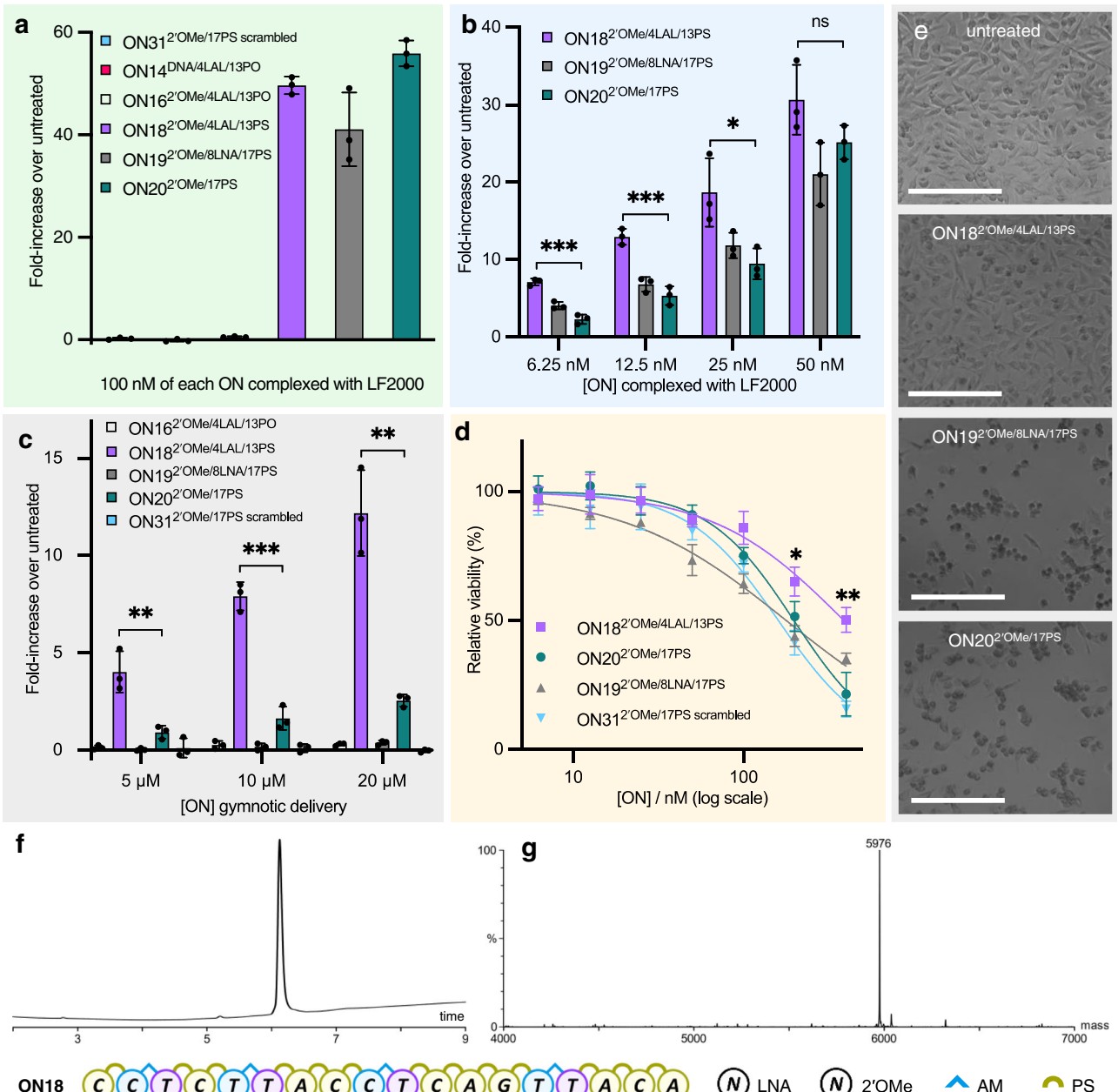

**Fig. 6 LNA-flanked amides and PS linkages increase gymnotic delivery, exon-skipping activity and reduce toxicity.** In all cases, luciferase activity was measured and normalised to both protein quantity and untreated cells. Experimental details and sequences are given in the Supplementary Information. **a** HeLa pLuc/705 cells were transfected at 100 nM ON using LF2000. **b** Dose-response for active ONs in **a**. **c** Cells were treated with ONs without a transfection agent. **d** Viability of the HeLa pLuc/705 cells following lipofection with ONs using LF2000 determined using WST-1 assay. For all graphs (**a–d**) data were means ± SD and values for ON18[2'OMe/4LAL/13PS] and ON20[2'OMe/17PS] were compared using unpaired two-sided t-test analysis. ns represents $P > 0.05$, *represents $P < 0.05$, **represents $P < 0.01$, ***represents $P < 0.001$. Each dot represents one distinct replicate ($n = 3$). ON31[2'OMe/17PS scrambled]: CCUCAUUCACUCGAUUCA. **e** Change in cell viability after treatment with 200 nM ONs complexed with LF2000 compared to untreated, scale bar = 200 μm. Micrographs of cells shown are the representations from ≥10 images taken per condition. **f** Reversed-phase HPLC chromatogram and **g**. Electrospray (ES-) mass spectrum of purified ON18 used in the assays (required mass = 5976.2, found mass = 5975.5). Source data are provided in the Source Data file.

LNA-amides and interactions of the PS backbone with cellular components. Improved cell uptake may result from the neutral amide linkages breaking up the poly-anionic backbone into short segments which penetrate cells more readily than long poly-anionic stretches. Interestingly, ON19[2'OMe/8LNA/PS] with LNA and no amides showed only a slight dose-response in activity, even at the highest concentration tested (Fig. 6c and Supplementary Fig. 24). This could be due to its binding to off-targets, altered rigidity, or

undesirable secondary structures induced by the extreme stability caused by the LNA sugars, reducing the ability of the ON to interact with the cell surface, a mechanism for productive uptake. ON16[2'OMe/4LAL/13PO] with LNA-amides and no PS linkages also displayed slight gymnotic splice-switching activity (Supplementary Fig. 24).

We compared the viability of the HeLa cells following lipofection using a WST-1 cell proliferation assay (Fig. 6d). At the highest concentration tested (400 nM) the cells treated with ON20[2'OMe/17PS]

were only 21% viable, whereas the cells treated with the same concentration of ON18$^{2'OMe/4LAL/13PS}$ were 50% viable, demonstrating that the LNA-amide linkage significantly reduces the cytotoxicity of ONs delivered with LF2000. This is verified by analysis of the protein levels (Supplementary Fig. 25) and visible cell death (Fig. 6e and Supplementary Fig. 26). It supports the use of the combination of LNA-amide, 2′-OMe and PS modifications for in vitro studies. Interestingly, the oligonucleotide containing eight LNA sugars without amide linkages (ON19$^{2'OMe/8LNA/17PS}$) had a poor toxicity profile. This could be due to off-target effects and could also explain why, despite showing the highest affinity towards RNA in the UV-melting studies, the LNA modified ON19$^{2'OMe/8LNA/17PS}$ was not the most active in the exon-skipping assay. Further detailed studies are required to determine whether this is a general or a sequence-specific phenomenon, and to validate its relevance in terms of toxicity.

Given that cell uptake and toxicity remain major challenges when developing new therapeutic oligonucleotides, the results in Fig. 6 suggest that our modification strategy could be advantageous. The synergistic effect of LNA-amide linkages and phosphorothioate modifications appear to produce oligonucleotides with enhanced biological properties. However, cell culture studies cannot address some of the major challenges in oligonucleotide therapeutics, particularly those that relate to pharmacokinetics, pharmacodynamics, biodistribution and aspects of toxicity. We are therefore planning further detailed biological studies on LNA-amide-phosphorothioate oligonucleotides to answer these important questions.

We have developed a high-yielding methodology to synthesise oligonucleotides containing uncharged LNA-amide linkages. The chemistry has the potential to be automated and carried out at scale for therapeutic oligonucleotide development. These new constructs have high resistance to enzymatic degradation and bind to complementary RNA with affinity and selectivity superior to unmodified ONs. The artificial backbone causes minimal structural deviation in DNA:RNA hybrids, consistent with their strong affinity for RNA. Oligonucleotides with alternating LNA-amide and phosphodiester (or phosphorothioate) backbones cannot give rise to recyclable LNA mononucleotides (modified dNTPs) in the presence of cellular nucleases, and their favourable toxicity profile relative to LNA in these initial studies may reflect this. In studies with gymnotic (naked) delivery, combining LNA-amides with phosphorothioates improves cell uptake. Poor cellular uptake is currently a major barrier in oligonucleotide therapeutics and combining the PS and LNA modifications with charge-neutral amide backbones such as AM1 could lead to improved clinical efficacy. Research is in progress to explore these new analogues in a range of cellular assays and in other therapeutic interventions such as siRNA and RNase-H mediated antisense inhibition. Finally, artificial nucleic backbones are of interest to researchers in many other fields including nucleic acid chemistry, chemical biology, biochemistry, medicinal chemistry, diagnostics, gene synthesis, gene editing, nanotechnology, materials chemistry and biophysics. We hope that this work will catalyse future research in these areas.

## Methods
The sequences of the oligonucleotides used in this study are provided in Supplementary Table 1.

### Amide-modified oligonucleotide synthesis
*Oligonucleotide segment synthesis.* Oligonucleotide synthesis was performed on an Applied Biosystems 394 automated DNA/RNA synthesiser on a 1.0-μmole scale using a standard phosphoramidite cycle of detritylation, coupling, and oxidation. No capping step was used. All β-cyanoethyl phosphoramidite monomers were dissolved in anhydrous MeCN (10% CH$_2$Cl$_2$ was added when 2′-OMe U phosphoramidite was used) to a concentration of 0.1 M immediately prior to use. 5-(Benzylthio)-1*H*-tetrazole (BTT) activator (0.3 M) was used with a coupling time of 50 s for normal dA, dG, dC and T phosphoramidites, this was extended to 6 min

for 2′-OMe and LNA phosphoramidites. Standard iodine oxidiser was used for phosphodiester oligonucleotides. For the phosphorothioate oligonucleotides, 3-ethoxy-1,2,4-dithiazoline-5-one (EDITH) was used as the sulfurisation agent, and the solid support was washed with MeCN after each phosphoramidite coupling before the sulfurisation step. Sulfurisation time was initially 3 min, and after this period fresh EDITH was sent to the synthesis column and left for another 3 min. Further details are given in the Supplementary Information.

*Amino monomer addition.* The MMT-protected 5′-amino phosphoramidite monomer (either LNA 10[47] or commercially available deoxythymidyl 11) was dissolved in anhydrous MeCN to a concentration of 0.1 M immediately prior to use. The same coupling conditions as above were used, but the coupling time was extended to 10 min. The MMT protecting group was cleaved on the Applied Biosystems 394 automated synthesiser using 3% TCA in CH$_2$Cl$_2$ with an extended cleavage time of 2 min. The solid support was then washed with acetonitrile on the synthesiser for 3 min. To improve the coupling efficiency in the next step, the solid support was washed with 0.5% (v/v) N-methylmorpholine in DMF (1 × 1 mL) followed by DMF (3 × 1 mL).

*Amide bond formation on the resin (peptide coupling).* All amide couplings were performed manually on the synthesis column. A solution with 10 equivalents of acid monomer, 10 equivalents of PyBOP and 30 equivalents of N-methylmorpholine was first prepared in 400 μL of DMF. This was then taken up into a 1 mL syringe and loaded onto the column before a second 1 mL syringe was attached to the other end of the synthesis column. The mixture was agitated every 10 min for 1 h. The columns were then washed with DMF (3 × 1 mL) followed by MeCN (5 × 1 mL) and dried by passing argon through the column. The column was then returned to the synthesiser to continue oligonucleotide synthesis.

*Cleavage of oligonucleotides from resin, deprotection and purification.* LNA-amide-containing oligonucleotides were isolated with the final 5′-DMT protecting group still in place (DMT-ON). Following solid-phase synthesis, the cyanoethyl groups were removed by a 15 min treatment with 20% diethylamine in MeCN. The resin was then washed with MeCN (5 × 1 mL) and dried by passing a stream of argon through the synthesis column. The oligonucleotides were cleaved from the solid support and deprotected by heating in concentrated aqueous ammonia in a sealed glass vial at 55 °C for 5 h. The ammonia was removed under reduced pressure prior to oligonucleotide purification. The DMT-ON oligonucleotides were purified by reverse-phase high-performance liquid chromatography (RP-HPLC) and lyophilised. They were then dissolved in 0.5 mL of 80% acetic acid and left for 1 hour at room temperature to remove the DMT group. The solution was neutralised with 0.5 mL of triethylammonium acetate buffer (2 M, pH 7) and the detritylated oligonucleotides were desalted using a NAP-10 column (Cytiva) and then freeze-dried.

**Reporting summary.** Further information on research design is available in the Nature Research Reporting Summary linked to this article.

## Data availability
The X-ray crystallographic coordinates for structures reported in this study have been deposited at the Cambridge Crystallographic Data Centre (CCDC), under deposition numbers 2105684 and 2105685. These data can be obtained free of charge from The Cambridge Crystallographic Data Centre via www.ccdc.cam.ac.uk/data_request/cif. The DNA structural data obtained by X-ray crystallography have been deposited in the Protein Data Bank (PDB) and are available with the following accession codes 7NRP, 7OOS, 7OZZ and 7OOO. The authors declare that all other data supporting the findings of this study are available within the paper and its supplementary information files. Data were available from the corresponding author upon request. Source data are provided with this paper.

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

## Acknowledgements

The authors would like to thank Dr Samir El-Andaloussi for the HeLa pLuc/705 cell line[52]. This work was funded in part by a BBSRC research grant to T.B. (New oligonucleotide analogues for therapeutic applications. BB/S018794/1).

## Author contributions

Y.R.B., L.M.P., L.L. and P.K performed the small molecule synthesis; Y.R.B., L.M.P. and K.E.C. carried out the small molecule crystallography work; Y.R.B., A.H.E.-S., J.C., S.E. and D.S. synthesised and purified the ONs; Y.R.B., J.C. and L.C. performed UV-melting and enzymatic stability studies; C.T., M.A.McD., J.S.H. and J.P.H. carried out the ON XRD studies; Y.R.B., W.F.L. and G.Mc.C. performed the cell studies; Y.R.B., A.H.E.L-S, P.K. and T.B. designed the study. All authors contributed to the writing of the manuscript. T.B. oversaw and managed the project. M.J.A.W. oversaw the biological work.

## Competing interests

T.B. is a consultant to ATDBio Ltd. The remaining authors declare no competing interests.
