## [Peer Review File · Nature Communications]

An LNA-amide modification that enhances the cell uptake and activity of phosphorothioate exon-skipping oligonucleotidesEditorial Note: This manuscript has been previously reviewed at another journal that is not operating a transparent peer review scheme. This document only contains reviewer comments and rebuttal letters for versions considered at *Nature Communications*.

REVIEWER COMMENTS

Reviewer #1 (Remarks to the Author):

The authors have submitted a revised version of their manuscript that addresses most of the concerns raised by reviewers in round 1.

Overall, the strengths of this paper are the novel chemical analogue (amide-bridged LNA) and the surprising finding that this analogue shows better binding affinity than previous amide-based mimics of the phosphate linkage. The authors show crystallographic work to rationalize the high binding affinity, and demonstrate promising preliminary gene silencing data in cells.

Without in vivo data, it is hard to know whether this modification will become a leading contender for therapeutic applications, and I strongly encourage the authors to explore this in future work. But already the paper includes lots of data reflecting a significant synthetic achievement, and an unexpectedly high binding affinity that amply merits publication in *Nature Communications*.

I have no further changes to request at this round of review.

Reviewer #2 (Remarks to the Author):

Unfortunately, this manuscript is minimally responsive to the previous review.

In my previous review, I suggested that the authors tone down their conclusions and do a few more experiments. I cannot tell whether the conclusions are toned down, because changes were not marked. The Title is not incorrect - any new chemistry has therapeutic potential - but it also does not characterize the value of the paper accurately. The most disturbing deficiency is the failure to add additional data on knocking down gene expression. If this work were a real advance, shouldn't it be easy to do these experiments? There are many active ASO sequences available for comparison, all of them with known and straightforward assays. Why should this reviewer believe that "this work will catalyze future research in these areas" if the authors cannot do a few simple experiments looking at well characterized gene targets? If the synthesis is too difficult, just say so and speculate on what will be needed to make extensive cell culture work practical.

While poor cellular uptake is a problem, much of that involves biodistribution constraints for many organs. It is not clear how this approach will help and the data don't support that these modification will help.

I have no problem with this work being published as an exercise in interesting chemistry. New chemistries are wonderful and no apologies need be made regarding chemistry for the sake of chemistry. However, as I outlined in my previous review, it should not be sold as something that it is not. Novel chemistry should be justified in terms of novel chemistry.

Reviewer #3 (Remarks to the Author):

Therapeutic oligonucleotides have shown great promise as precision medicines, and have been approved in clinical use. The artificial nucleic acids have been developed to improve the enzymatic

resistance, cellular uptake efficiency, and toxicity of therapeutic oligonucleotides. This manuscript reported the design and synthesis of charge-neutral locked nucleic acid-amide linkages, and incorporate them into the natural DNA to obtain the reduced charge oligonucleotides. The artificial neutral linkage significantly improved the improved cell uptake efficiency, RNA affinity and stability, providing the researchers the new tools in gene therapy.

We approved the ingenious work in molecular design and the structure analysis of chimeric duplex, and believe this work would contribute heavily to gene therapy. We do believe this manuscript should be published in Nature Communication.

Reviewer #4 (Remarks to the Author):

The manuscript "Oligonucleotide analogues with locked-amide linkages have therapeutic potential" was submitted by Tom Brown et al is nice interesting work based on the application of oligonucleotides for clinical use in the treatment of chronic diseases. Here are some points which should be answered before final acceptance of the manuscript is granted

1. Which chronic diseases are these therapeutic molecules targeted? please specify.
2. The authors have mentioned the poor toxicity of these compounds ., I suggest some genotoxicity viz, Micronucleus Test MT and comet assay studies would add value to the oligonucleotides therapeutics.
3. Comparative DNA and RNA studies should be carried out to validate the binding affinity of these compounds and that would confirm the RNA targeting.

Overall the methodology used by the authors seems quite rational and sound

Reviewer #5 (Remarks to the Author):

As requested by the editor I had a careful look into the X-ray structure analysis of the amide-linked duplexes. The X-ray data collection and crystallographic refinement statistics are fine. My recommendation is to provide a Supporting Figure showing the $2F_{obs} - F_{calc}$ electron density map (contoured at 1.0 or 1.5 sigma level) for the amide linkage.

RESPONSE TO REVIEWER COMMENTS

Reviewer #1 (Remarks to the Author):

The authors have submitted a revised version of their manuscript that addresses most of the concerns raised by reviewers in round 1.

Response: We thank the reviewer for confirming this.

Overall, the strengths of this paper are the novel chemical analogue (amide-bridged LNA) and the surprising finding that this analogue shows better binding affinity than previous amide-based mimics of the phosphate linkage. The authors show crystallographic work to rationalize the high binding affinity, and demonstrate promising preliminary gene silencing data in cells.

Response: We thank the reviewer for confirming this. We appreciate that the reviewer acknowledges that the cell work is preliminary. A minor point is that the cell studies are on exon-skipping (restoration of function) rather than gene silencing.

Without in vivo data, it is hard to know whether this modification will become a leading contender for therapeutic applications, and I strongly encourage the authors to explore this in future work. But already the paper includes lots of data reflecting a significant synthetic achievement, and an unexpectedly high binding affinity that amply merits publication in Nature Communications.

Response: We thank the reviewer for their positive comments on the content of the paper. We agree that the modification shows potential and warrants further work. We will carry out a thorough study to investigate the PK/PD, toxicity, and biodistribution for which we have recently obtained funding (commencing in August 2022).

I have no further changes to request at this round of review.

Response: We thank the reviewer for their supportive comments.

Reviewer #2 (Remarks to the Author):

Unfortunately, this manuscript is minimally responsive to the previous review. In my previous review, I suggested that the authors tone down their conclusions and do a few more experiments. I cannot tell whether the conclusions are toned down, because changes were not marked.

Response: The conclusions were toned down in the revised version of the paper. We are sorry that the reviewer did not receive a marked version of the manuscript.

The Title is not incorrect - any new chemistry has therapeutic potential - but it also does not characterize the value of the paper accurately.

Response: Our use of the word potential was to suggest the modifications warrant further investigation in the context of nucleic acid therapeutics. We apologise if it might suggest that we have developed a new therapeutic. We have changed the title to:

An LNA-amide modification that enhances the cell uptake and activity of phosphorothioate exon-skipping oligonucleotides.

We do not agree with the reviewers point that any new chemistry has therapeutic potential. Hundreds of different nucleotide analogues and phosphate mimics have been prepared and reported in the literature. However, only a handful of these are beneficial in terms of RNA binding (which is essential to achieve a therapeutic effect) and ability to recognise mismatches in the target RNA (necessary to avoid off target effects). Even fewer have been shown to improve gymnotic delivery. In such an interdisciplinary area as nucleic acid research, it is understandable that experts in the field may not appreciate the challenges faced in chemical development, but we ask that the reviewer views Figure 6 as a small but important part of a much larger paper. We wish to encourage those with the expertise to evaluate novel modifications, but without the necessary chemical background, to study this modification further, and also encourage others to consider charge neutral backbones combined with phosphorothioate linkages.

The most disturbing deficiency is the failure to add additional data on knocking down gene expression. If this work were a real advance, shouldn't it be easy to do these experiments? There are many active ASO sequences available for comparison, all of them with known and straightforward assays. Why should this reviewer believes that "this work will catalyze future research in these areas" if the authors cannot do a few simple experiments looking at well characterized gene targets? If the synthesis is too difficult, just say so and speculate on what will be needed to make extensive cell culture work practical.

Response: As discussed above, developing chemical modifications that can increase the binding affinity of neutral oligonucleotide backbones represents a large body of work. We feel that further biological studies are beyond the scope of the current manuscript and, if done in sufficient depth, would require a second manuscript. The current manuscript already contains a substantial number of diverse experiments, and we would rather carry out further biological studies in future. To study gene-knockdown, as suggested by the reviewer, would also be a diversion; in this paper we carry out exon-skipping to restore gene function, not knock down of gene expression. In the context of knock-down of gene expression, we have previously shown that the DNA-amide linkage (i.e. not LNA-amide) can be productively placed in the wings of gapmer ASOs, but affects RNase-H activity when placed within the gap:

Consecutive 5'- and 3'-amide linkages stabilise antisense oligonucleotides and elicit an efficient RNase H response S. Eppler, C. Thorpe, Y. R. Baker, A. H. El-Sagheer and T. Brown. Chem. Comm. 2020 56 (41), 5496-5499. DOI: <https://doi.org/10.1039/D0CC00444H>

To study RNase-H-mediated gene knockdown with LNA-amide, we would need to make a series of oligos where we walk single and multiple LNA-amide linkages through the oligonucleotide to determine if they (i) help oligonucleotide delivery, (ii) reduce cellular toxicity and (iii) are tolerated by RNase-H at specific positions in the gap. This would require a large amount of chemical synthesis, *in vitro* and cell experiments, and an additional detailed discussion in the paper. This is an example of what would seem to be a few simple experiments leading to a large volume of work.

While poor cellular uptake is a problem, much of that involves biodistribution constraints for

many organs. It is not clear how this approach will help and the data don't support that these modification will help.

Response: We agree that cell culture studies cannot address one of the major challenges in oligonucleotide therapeutics, namely issues related to pharmacokinetics, biodistribution, organ toxicity, and clearance. We have recently obtained funding to carry out a detailed biological study on these factors which will involve animal models. The start date is August 2022, and we hope to obtain a substantial amount of data that will form the basis of a publication addressing these important issues that the reviewer has highlighted.

Given the concerns of the reviewer, we have made a further change to the paper to explain the importance of future PK studies at the end of the section on cell culture assays:

“However, cell culture studies cannot address some of the major challenges in oligonucleotide therapeutics, particularly those that relate to pharmacokinetics, pharmacodynamics, biodistribution and aspects of toxicity. We are therefore planning further detailed biological studies on LNA-amide-phosphorothioate oligonucleotides to answer these important questions.”

I have no problem with this work being published as an exercise in interesting chemistry. New chemistries are wonderful and no apologies need be made regarding chemistry for the sake of chemistry. However, as I outlined in my previous review, it should not be sold as something that it is not. Novel chemistry should be justified in terms of novel chemistry.

Response: We thank the reviewer for these comments.

Reviewer #3 (Remarks to the Author):

Therapeutic oligonucleotides have shown great promise as precision medicines, and have been approved in clinical use. The artificial nucleic acids have been developed to improve the enzymatic resistance, cellular uptake efficiency, and toxicity of therapeutic oligonucleotides. This manuscript reported the design and synthesis of charge-neutral locked nucleic acid-amide linkages, and incorporate them into the natural DNA to obtain the reduced charge oligonucleotides. The artificial neutral linkage significantly improved the improved cell uptake efficiency, RNA affinity and stability, providing the researchers the new tools in gene therapy.

We approved the ingenious work in molecular design and the structure analysis of chimeric duplex, and believe this work would contribute heavily to gene therapy. We do believe this manuscript should be published in Nature Communication.

Response: We thank the reviewer for their supportive comments

Reviewer #4 (Remarks to the Author):

The manuscript "Oligonucleotide analogues with locked-amide linkages have therapeutic potential" was submitted by Tom Brown et al is nice interesting work based on the application of oligonucleotides for clinical use in the treatment of chronic diseases. Here are some points which should be answered before final acceptance of the manuscript is granted

1. Which chronic diseases are these therapeutic molecules targeted? please specify.

Response: They can be used to treat a very wide range of diseases which can be addressed by targeting RNA, as discussed in the introduction. There is insufficient space available in the

manuscript to discuss this in more detail, but the subject is covered in-depth in references 1 - 12.

2. The authors have mentioned the poor toxicity of these compounds ., I suggest some genotoxicity viz, Micronucleus Test MT and comet assay studies would add value to the oligonucleotides therapeutics.

Response: We agree with reviewer 3 that potential toxicity warrants further investigation, However, we feel that these additional studies are beyond the scope of the current manuscript. We have recently obtained funding to carry out detailed biological studies on our LNA-amide-phosphorothioate oligonucleotides involving further cell and animal work, and we anticipate that this will provide a substantial amount of new data for publication. An additional consideration is that the toxicity assays suggested by the reviewer look at genotoxicity which is not thought to be a problem for phosphorothioate or LNA modifications. Given that our LNA-amide modification appears to be less toxic than the parent oligos, we are doubtful that we would be able to observe genotoxicity at a quantifiable level.

3. Comparative DNA and RNA studies should be carried out to validate the binding affinity of these compounds and that would confirm the RNA targeting.

Response: Our modified oligonucleotides are indeed selective for RNA over DNA targets. Comparative DNA and RNA binding affinity studies are described in the paper (pages 10, 11) where we state:

“In duplexes with DNA targets, ONs with all combinations of LNA and DNA sugars around the amide linkage were very slightly destabilising (between -0.1 °C to -2.6 °C), indicating the selectivity of the amide linkage for complementary RNA.”

This covers RNA and DNA affinity using UV melting (the standard technique used in the field to compare RNA binding/targeting) and also shows that base pair mismatches strongly reduce RNA affinity. Data are provided in the supporting information (Figures S8, S14, S16) and in supplementary Table 2.

Overall the methodology used by the authors seems quite rational and sound

Response: We thank the reviewer for their supportive comments.

Reviewer #5 (Remarks to the Author):

As requested by the editor I had a careful look into the X-ray structure analysis of the amide-linked duplexes. The X-ray data collection and crystallographic refinement statistics are fine. My recommendation is to provide a Supporting Figure showing the $2F_o - F_c$ electron density map (contoured at 1.0 or 1.5 sigma level) for the amide linkage.

Response: This is a good point. We have added a $2F_o - F_c$ electron density map at the 1 σ contour level for all four nucleic acid crystal structures reported in the manuscript. This is now Figure S20 in the revised supporting information. We have also added the following sentence to the paper:

“Electron density maps at the modification position for the four nucleic acid crystal structures reported in this manuscript are given in Supplementary Fig. 20.”

We thank the reviewer for confirming that the X-ray data collection and crystallographic refinement statistics are fine.

Other suggested changes

In Figure 1, we use a traffic light system to indicate the advantages and disadvantages of existing nucleic acid modifications with respect to their therapeutic suitability. This is a simple graphic system to represent a complex issue. In this system AM1 could be equally represented as amber for duplex stability, and LNA amber for toxicity instead of red. PMO could possibly be considered as green for toxicity and the 2'-ribose modifications fall between red and amber for cell uptake. We have made these changes, and changed the symbols for the benefit of colour-blind readers. Similarly, we have changed the green and purple melting curves in Figure 4 from solid lines to dotted lines.

None of the referees commented on the above.

We have also made some minor stylistic changes to fit with Journal policy, including shortening sub-titles. All changes in the manuscript are marked in red.

Source data is provided for the charts in Figure 6 and Supplementary Figures 24 and 25.

REVIEWERS' COMMENTS

Reviewer #5 (Remarks to the Author):

As suggested the authors added a Supporting Figure showing the electron density at the amide linkage, supporting the high quality of their X-ray analysis. From my side, I support acceptance of the manuscript..